**PLOS** COMPUTATIONAL BIOLOGY

# Use of Angiotensin-Converting Enzyme Inhibitors and Angiotensin II Receptor Blockers During the COVID-19 Pandemic: A Modeling Analysis

Mehrshad Sadria[1], Anita T. Layton [1,2]*

**1** Department of Applied Mathematics, University of Waterloo, Waterloo, Ontario, Canada, **2** Department of Biology, Cheriton School of Computer Science, and School of Pharmacy, University of Waterloo, Waterloo, Ontario, Canada

\* anita.layton@uwaterloo.ca

**Data Availability Statement:** All relevant data are within the manuscript and its Supporting Information files.

## Abstract

Angiotensin-converting enzyme inhibitors (ACEi) and angiotensin II receptor blockers (ARB) are frequently prescribed for a range of diseases including hypertension, proteinuric chronic kidney disease, and heart failure. There is evidence indicating that these drugs upregulate ACE2, a key component of the renin-angiotensin system (RAS) and is found on the cells of a number of tissues, including the epithelial cells in the lungs. While ACE2 has a beneficial role in many diseases such as hypertension, diabetes, and cardiovascular disease, it also serves as a receptor for both SARS-CoV and SARS-CoV-2 via binding with the spike protein of the virus, thereby allowing it entry into host cells. Thus, it has been suggested that these therapies can theoretically increase the risk of SARS-CoV-2 infection and cause more severe COVID-19. Given the success of ACEi and ARBs in cardiovascular diseases, we seek to gain insights into the implications of these medications in the pathogenesis of COVID-19. To that end, we have developed a mathematical model that represents the RAS, binding of ACE2 with SARS-CoV-2 and the subsequent cell entry, and the host's acute inflammatory response. The model can simulate different levels of SARS-CoV-2 exposure, and represent the effect of commonly prescribed anti-hypertensive medications, ACEi and ARB, and predict tissue damage. Model simulations indicate that whether the extent of tissue damage may be exacerbated by ACEi or ARB treatment depends on a number of factors, including the level of existing inflammation, dosage, and the effect of the drugs on ACE2 protein abundance. The findings of this study can serve as the first step in the development of appropriate and more comprehensive guidelines for the prescription of ACEi and ARB in the current and future coronavirus pandemics.

## Author summary

As we brace for the devastating impact of the COVID-19 pandemic, we must tackle a controversy on how to best minimize the risk of lethal disease among the most vulnerable.

**Funding:** This research was supported by the Canada 150 Research Chair program (A.T.L.) and the NSERC Discovery award (A.T.L.). The funders had no role in study design, data collection and analysis, decision to publish, or preparation of the manuscript.

**Competing interests:** The authors have declared that no competing interests exist.

Preliminary epidemiological data show an exponential increase in disease severity and mortality among patients with cardiovascular disease and diabetes. The coronavirus enters host cells by binding to a specific enzyme "ACE2" on the cell membrane. ACE2 abundance is increased in patients with cardiovascular disease and diabetes treated with two classes of drugs: angiotensin-converting enzyme inhibitors (ACEi) and angiotensin II receptor blockers (ARB). This has generated controversy regarding the approach for patients taking these drugs during the pandemic, with some advocating for discontinuing these medications, while expert opinions recommended against discontinuation, given the lack of strong evidence. Given the success of ACEi and ARBs in cardiovascular diseases, we aim to help patients and physicians weigh the overall pros and cons. To achieve that goal, we have developed a mathematical model of the invasion of the coronavirus and the host's immune response. Model simulations indicate how much tissue damage COVID-19 induces in a patient undergoing ACEi or ARB treatment depends on a number of factors, including the level of any existing chronic inflammation, dosage, and certain drugs effects.

## Introduction

The Severe Acute Respiratory Syndrome Coronavirus 2 (SARS-CoV-2) has wreaked havoc all over the world. SARS-CoV-2 has a higher transmission rate than the SARS-CoV from 2003, and causes the coronavirus disease 2019 (COVID-19). In some patients COVID-19 causes acute respiratory distress syndrome, which has high morbidity and mortality. There appears to be some indications that COVID-19 tends to be more severe in patients with hypertension and diabetes [1], although it is difficult to assess to what extent that preliminary conclusion can be attributable to bias in age, sex, comorbidities, and existing medication. Nonetheless, concern has emerged that some anti-hypertensive treatments, specifically renin-angiotensin system (RAS) inhibitors, may increase the risk of SARS-CoV-2 infection and lead to more severe COVID-19 owing to the RAS-mediated cell entry mechanism of the virus [2].

Both SARS-CoV and SARS-CoV-2 gain entry into host cells via the binding of its spike protein with the angiotensin converting enzyme 2 (ACE2). ACE2 is a key component of the RAS and is found on the cells of a number of tissues, including the type 2 alveolar epithelial cells in the lungs [3]. The actions of ACE2, and its homologue ACE, are described below. Because ACE2 opposes the actions of the vasoactive angiotensin II (Ang II), ACE2 has a beneficial role in many diseases such as hypertension, diabetes, and cardiovascular disease in which its expression is decreased. Following binding of its spike protein to ACE2 and proteolytic cleavage of ACE2 by transmembrane serine protease 2 (TMPRSS2), the virus enters the cell and replicates.

The RAS consists of an enzymatic cascade that leads to the production of Ang II [4]. The cascade starts with angiotensinogen, which is cleaved by renin, followed by angiotensin converting enzyme (ACE) and neutral endopeptidase (NEP), to yield different forms of angiotensin (see Fig 1) [4–6]. The first reaction in the RAS is catalyzed by renin, which is often considered the driving force in the cascade. The major products of the RAS include angiotensin (1–7) (Ang (1–7)) and Ang II, which bind to receptors and exert their effects on the brain, heart, kidney, vasculature, and immune system [4]. Ang II binds to two different types of receptors: Ang II type 1 receptor (AT1R) and type 2 receptor (AT2R). When bound to AT1R, Ang II stimulates renal vasoconstriction, promotes sodium reabsorption, inflammation, and fibrosis [5]. In contrast, when bound to AT2R, Ang II induces vasodilation and natriuresis.

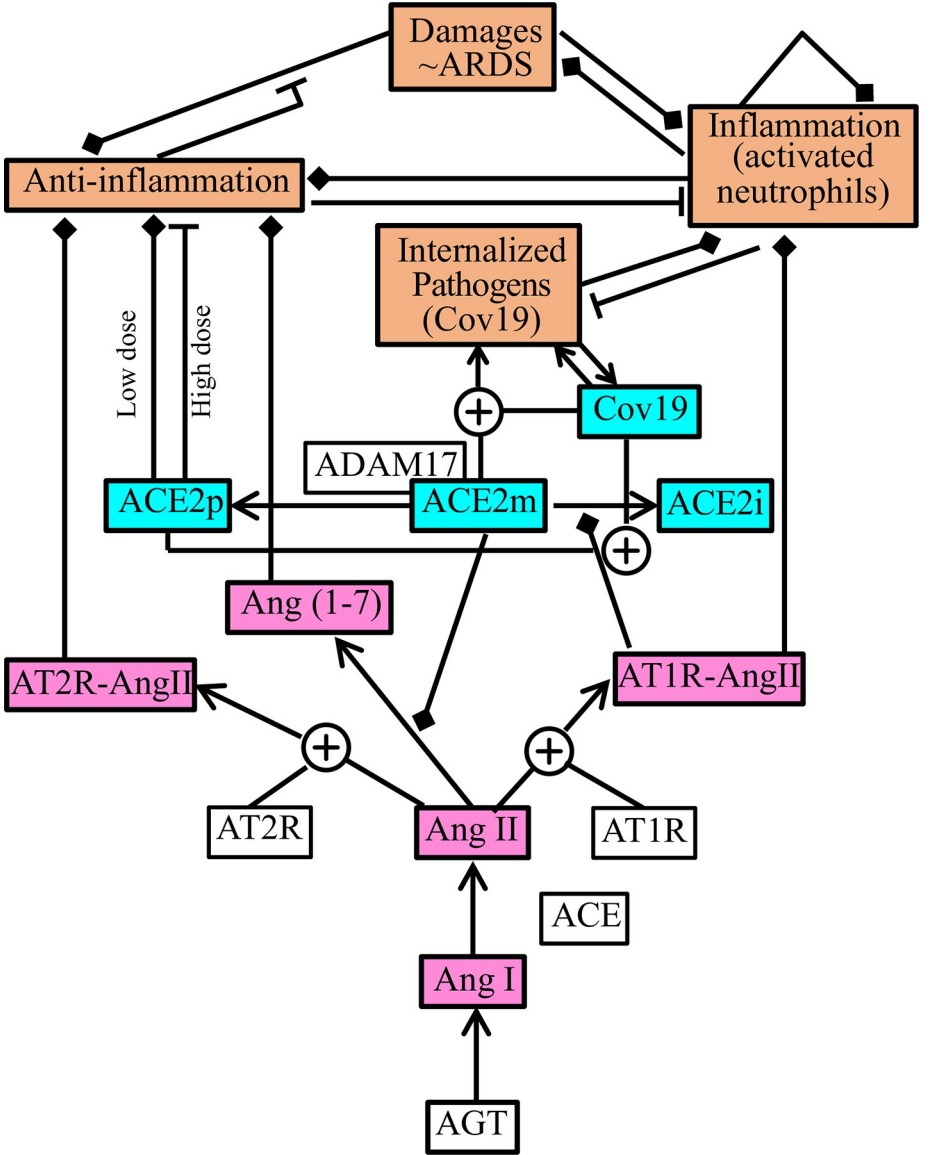

**Fig 1. Schematic model diagram.** Orange boxes, inflammatory response; blue boxes, angiotensin converting enzyme 2 (ACE2) dynamics; purple boxes, renin-angiotensin system; white boxes, model parameters. Arrows represent conversion; square end, activation; flat end, inhibition.

When bound to the angiotensin type 7 receptor (AT7R), also known as the MAS receptor, Ang (1-7) induces vasodilation and natriuresis, elevates nitric oxide bioavailability, and attenuates inflammation and fibrosis [5, 6]. The conversion of Ang II to Ang (1-7) is promoted by ACE2. See Fig 1.

Due to the vasoactive and renal effects of AT1R-bound Ang II and AT2R-bound Ang II, the RAS is a target for blood pressure control. Indeed, some of the RAS inhibitors, such as ACE inhibitors (ACEi) and angiotensin II receptor blockers (ARBs), are prescribed to millions of patients worldwide. Because ACEi and ARBs are known to upregulate ACE2 mRNA level [7], it has been suggested that these therapies can theoretically increase the risk of SARS- CoV-2 infection and cause more severe COVID-19 owing to the role of ACE2 as the viral binding

site [2]. However, given the success of ACEi and ARBs in cardiovascular diseases, the decision to discontinue their treatment should not be made lightly. Hence, the goal of this study is to develop a mathematical model to gain insights into the effect of these drugs on the pathogenesis of COVID-19.

## Results

### Exposure level

We first consider the effect of differing levels of exposure on disease severity, i.e., quickly passing by an asymptomatic patient versus working alongside a symptomatic patient with poor personal hygiene. To simulate low versus high level of exposure, we initialize plasma SARS-CoV-2 to the corresponding level (1 versus 10 pathogen units). Key simulations are summarized in Fig 2, panels A1-A3 and B1-B3.

Plasma SARS-CoV-2 combines with membrane ACE2 to gain cell entry, increasing cellular SARAS-CoV-2 population (Fig 2A2 and 2B2) and causing tissue damage (Fig 2A1 and 2B1). When internalized SARS-CoV-2 exceeds a threshold ($P_c$), an immune response is triggered (Fig 2A3 and 2B3). At low exposure level, the immune response can suppress the viral population to a sufficiently low level for recovery (Fig 2A1). At sufficiently high exposure, however, tissue damage becomes severe and tissue recovery is not achieved (Fig 2B1). For a given set of

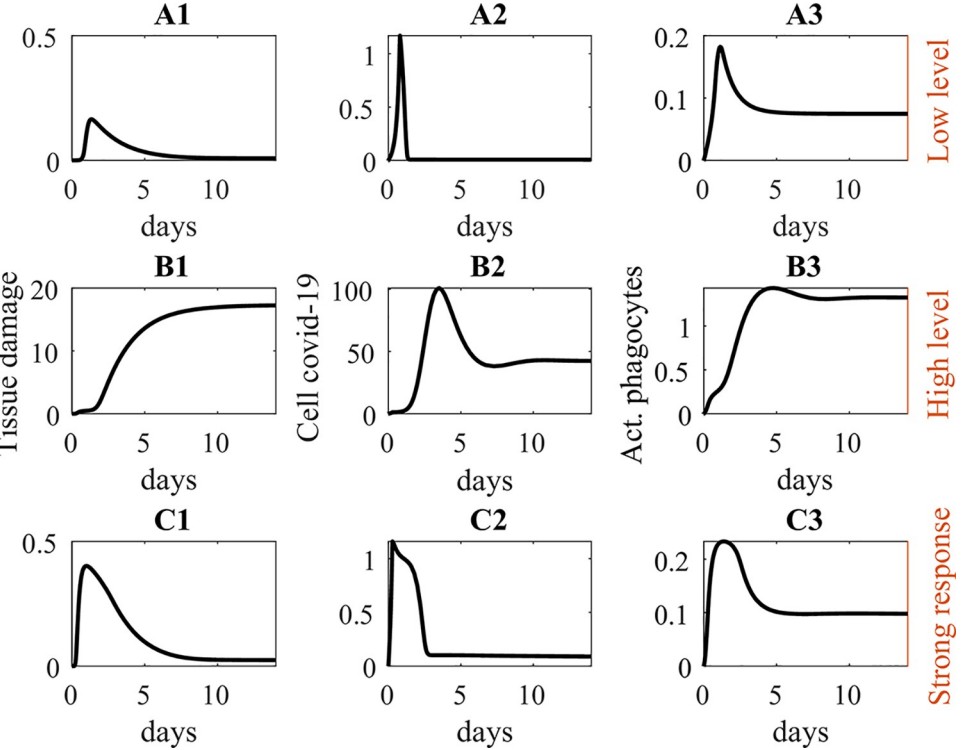

**Fig 2. Model response to low level initial exposure to covid-19 (panels A1-A3) versus high level (B1-B3), and effect of a stronger immune response with high exposure level (C1-C3).** Panels A1-C1, tissue damage (D). Recovery is possible at low exposure level (A1). High exposure level yields irrecoverable tissue damage (possibly death; B1), but recovery is possible if immune response is stronger (C1). Panels A2-C2, cellular SARS-CoV-2 level. Viral clearance is possible at low exposure level, or high level with a sufficiently strong immune response (A2 and C2), but clearance fails at high exposure level (B2). Panels A3-C3, inflammatory agents (e.g. activated phagocytes). Note different ordinate range for the B panels (nondimensional units).

parameters, the model prediction is interpreted as "tissue recovery" if the long-term tissue damage (D) approaches its initial small value or as "sustained tissue damage," when a sufficiently large steady-state D value is predicted.

## Strength of immune response

We conduct additional sets of simulation in which we either (i) assume that an effective immune response is triggered earlier (i.e., at half the baseline internalized SARS-CoV-2 threshold $P_c$) or (ii) the immune response, once triggered, is more effective in reducing the viral load (specifically, we double $k_{pm}$ and $k_{pn}$ in Eq 10 in METHODS; $k_{pm}$ and $k_{pn}$ characterize the strength of the local immune response and the inhibitory action of the activated phagocytes). In both cases, tissue recovery is predicted. Key results for the case with the stronger immune response is shown in Fig 2C1–2C3; results for the earlier trigger case are similar. These results highlight the importance of individual difference in immune response in determining disease outcome.

## Hypertension, ACEi, and ARBs

In the next set of simulations, we seek to understand the complex story around hypertension and some of its treatments in the context of COVID-19. Hypertension is a complex and multifactorial disease. We focus on the observation that low-grade chronic inflammation is often involved in the development and pathophysiology of hypertension [8]. Inflammatory status can be altered by a number of factors. For simplicity, we simulate an inflammatory state by increasing the generate rate of activated neutrophils ($s_{nr}$ in Eq 11 in METHODS).

**Existing inflammation may increase the severity of a COVID-19 infection.** We simulate the immune response of patients with an existing mild inflammation ($s_{nr} \times 2$) and with a more severe inflammation ($s_{nr} \times 5$). A low initial exposure level is assumed, so that the control (normotensive, without an existing inflammation) achieves tissue recovery (see Fig 2 and Table 1, "Normotensive" without drug treatment). The mild inflammation case attains tissue recovery as well (Table 1, "HTN mild inflame," without drug treatment). In contrast, the more

**Table 1. Simulated COVID-19 outcome for normotensive patients, hypertensive (HTN) patients with an existing mild inflammation, and HTN patients with a more severe inflammation.** ACEi, angiotensin-converting enzyme inhibitor ARB, angiotensin II receptor blocker. In the "ACEi" column, "Yes" corresponds to full effect of ACEi, i.e., reduced conversion rate from Ang I to Ang II and upregulation of ACE2; "w/o ACE2 ↑" and "ACE2 ↑ only 50%" represent only the reduced Ang I-to-Ang II conversion rate, with ACE2 level kept at baseline or 2.5-fold increase, respectively; "Half dose" represents half the ACEi effect, in both Ang I-to-Ang II conversion and ACE2 upregulation. Labels under the "ARB" column are defined analogously.

|  | ACEi | ARB | TISSUE Recovery? |
|---|---|---|---|
| Normotensive | — | — | Yes |
|  | Yes | — | No |
|  | w/o ACE2 ↑ | — | Yes |
|  | — | with or w/o ACE2 ↑ | No |
| HTN, mild inflam. | — | — | Yes |
|  | Yes | — | No |
|  | ACE2 ↑ only 50% | — | Yes |
|  | — | with or w/o ACE2 ↑ | No |
|  | Half dose | — | Yes |
|  | — | Half dose | Yes |
| HTN, severe inflam. | — | — | No |
|  | Yes | — | Yes |
|  | — | Yes | Yes |

severe inflammation case fails to reach tissue recovery, even with only low initial exposure. This result indicates that a sufficiently severe existing inflammation may increase the likelihood of serious complications following a COVID-19 infection.

**ACEi, ARBs and COVID-19.** As previously noted, concerns have been expressed regarding the possibility that ACEi may aggravate the severity of a SARS-CoV-2 infection, due to the upregulation of ACE2 [7], which facilitates SARS-CoV-2 cell entry, following the treatment [9]. The same concern may apply to ARBs, which have also been reported to upregulate ACE2 mRNA [7]. But given their major cardiovascular benefits, a premature rejection of these therapies might do more harm than good. Thus, we seek to glean insight into the complex story around ACEi, ARBs, and COVID-19. Specifically, we conduct simulations to determine whether the administration of ACEi and ARBs in different patient subpopulations changes tissue recovery outcomes. Our findings are summarized in Table 1 and Fig 3.

Following our previous work [10], ACE inhibition is modelled by reducing the conversion of Ang I to Ang II by 78%. Thus, ACE inhibition lowers Ang II, as well as AT1R-bound Ang II (pro-inflammatory) and AT2R-bound Ang II (anti-inflammatory) downstream. ACE inhibition is known to upregulate ACE2 mRNA level [7]. The higher abundance of ACE2 would increase (i) the rate of conversion of Ang II to Ang (1-7) which is anti-inflammatory, and (ii) the rate of SARS-CoV-2 cell entry. Both rates are increased by 5 folds based on the change in ACE2 mRNA level reported in Ref. [7], with the assumption that these rates are directly proportional to ACE2 mRNA level. Similarly, ARB is modelled by reducing the conversion of Ang II to AT1R-bound Ang II by 67% [10]. Taken in isolation, ARB does not affect Ang II, but lowers the amount of AT1R-bound Ang II (pro-inflammatory) and increases the amount of AT2R-bound Ang II (anti-inflammatory). ACE2 abundance is assumed to increase 3 folds by ARB, again based on the change in ACE2 mRNA level [7].

**Normotension.** Consider a normotensive individual, without a significant existing inflammation. Following a low initial exposure to SARS-CoV-2, the model predicts tissue recovery in the absence of ACEi or ARBs. However, when ongoing ACEi treatment is simulated, tissue recovery is no longer possible following the same SARS-CoV-2 exposure. Why does ACEi impede tissue recovery? To understand the underlying mechanism, we eliminate

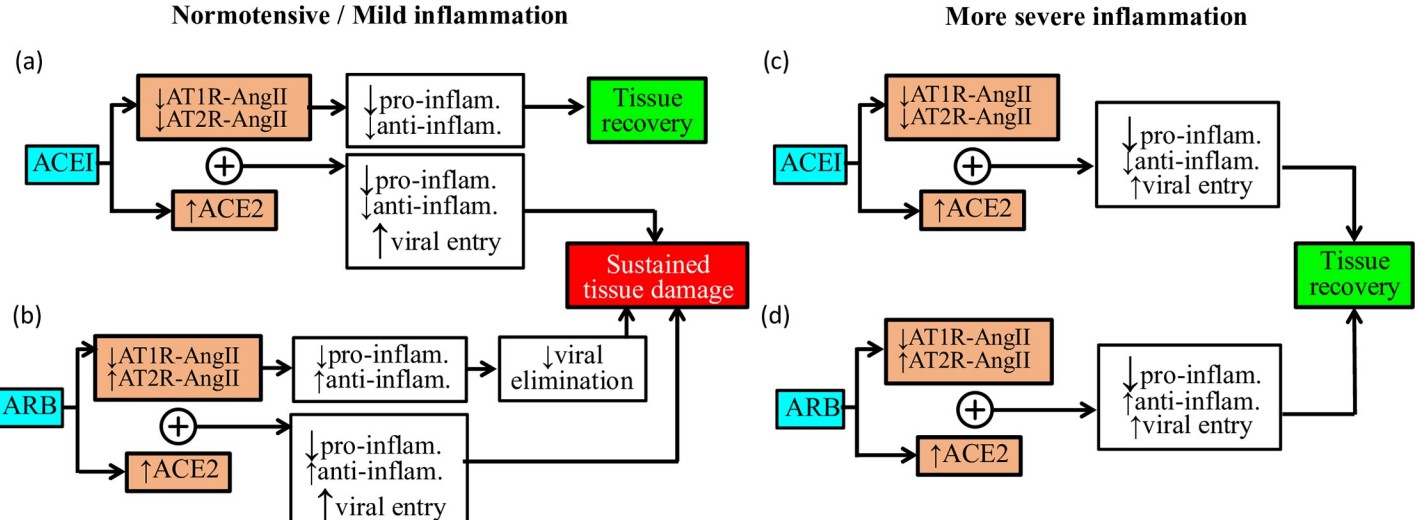

**Fig 3.** Schematic diagram summarizing predicted effects, and the underlying mechanisms, of ACEi (a and c) and ARB (b and d) under conditions of no or mild inflammation (a and b) or more severe inflammation (c and d).

the effect of ACEi on ACE2 abundance. That is, we only reduce the conversion of Ang I to Ang II by the same degree, but keep ACE2 abundance at baseline (specifically, we keep the rate of SARS-CoV-2 cell entry at baseline values). With this configuration, the mode predicts tissue recovery following SARS-CoV-2 exposure. This result suggests that, in this case, the upregulated ACE2 abundance, and the resulting higher SARS-CoV-2 cell entry rate, may be responsible, at least in part, for the potential negative impact of ACEi on tissue recovery in COVID-19. These results and the underlying mechanisms are summarized in Table 1 and Fig 3(A).

Consider the same normotensive individual, now under ARB treatment. The model again predicts sustained tissue damage. This time tissue recovery is not possible even with ACE2 abundance kept at baseline. By inhibiting the binding of Ang II to AT1R, ARB reduces the pro-inflammatory AT1R-bound Ang II (smaller $N^*$) while increasing the anti-inflammatory AT2R-bound Ang II (larger $C_A$). While these changes attenuate tissue damages the (the $k_{dn}f_s(f(N^*))$ term in Eq 12, METHODS), they also impede the defense against pathogens (the $k_{pn}f(N^*)P$ term in Eq 10). With the invasion of a new virus, attenuation of the inflammatory defense can be costly. (See Table 1 and Fig 3(A)). To confirm this line of reasoning, we conduct the same simulation, with the pro-inflammatory effect of AT1R-bound Ang II on $N^*$ enhanced (by increasing $k_{AT1R}$ 10 folds) and the anti-inflammatory effect of AT2R-bound Ang II on $C_A$ reduced (by decreasing $k_{AT2R}$ 10 folds). With these changes, the model predicts eventual recovery from COVID-19 induced tissue damage even under ARB treatment.

**Hypertension with mild inflammation.**   Next we consider a hypertensive patient, with an existing mild chronic inflammation. Following a low initial exposure to SARS-CoV-2, the model predicts tissue recovery. However, that tissue recovery is lost when combined with ACEi treatment. Again, why does ACEi impede tissue recovery? As in the normotensive case, the explanation lies with the ACEi-induced upregulation of ACE2. It is noteworthy that we assume 5-fold increase in ACE2, based on a study in rat cardiac tissues [7]. It is not clear to what extent the same increase should be found in human lung tissues. If the ACEi-induced ACE2 increase is lowered to 50%, then the model predicts tissue recovery. See Table 1 and Fig 3(a). This result suggests that the concern about the use of ACEi in patients with mild inflammation may depend on the drug's effect on ACE2 abundance, which has yet to be sufficiently well characterized in humans.

The model predicts that with ARB treatment, with or without ACE2 upregulation, tissue recovery is not possible (Table 1 and Fig 3(b)). The explanation is similar to the normotensive case above. It is noteworthy that the effects of ACEi and ARBs on the model's pro- and anti-inflammatory signals are not well quantified. Furthermore, our simulations indicate that when the effect of ACEi or ARB is halved (conversion of Ang I to Ang II reduced by 39% in ACEi and binding of Ang II to AT1R reduced by 33.5%), tissue recovery is attained. The attenuation of the effect of ACEi or ARB can be interpreted as a lower dosage. Thus, this result highlights the importance of dosage in determining the effects of ACEi and ARBs in the content of COVID-19.

**Hypertension with more severe inflammation.**   Finally, we consider a hypertensive patient with an existing more severe chronic inflammation. Unlike the previous cases, without ACEi or ARB treatment, sustained tissue damage is predicted following a low initial exposure to SARS-CoV-2. Interestingly, the model predicts that with the ACEi or ARB treatment, tissue recovery becomes possible. See Table 1 and (Fig 3(c) and 3(d)). It is important to note that this result is qualitatively different from the normotensive or mild inflammation cases. With an already elevated level of inflammation and tissue damage, the anti-inflammatory benefits of these drugs outweigh other factors, such as the increased ACE2-mediated cell entry by SARS-CoV-2. Thus, ACEi and ARBs may facilitate recovery from COVID-19 induced tissue damage in patients with an existing more severe chronic inflammation.

**Table 2. Parameter sensitivity study.** Percentage changes in peak viral load (P) and tissue damage (D) are reported only for cases that achieve tissue recovery.

| | | Δ max P (%) | Δ max D (%) | Tissue recovery? |
|---|---|---|---|---|
| $S_{ACE2m}$ | × 2 | — | — | No |
| | ÷ 2 | +5.7 | +133 | Yes |
| $c_{ADAM17}$ | × 2 | -5.0 | -20.5 | Yes |
| | ÷ 2 | — | — | No |
| $c_{AT1R}$ | × 2 | -1.3 | -6.4 | Yes |
| | ÷ 2 | — | — | No |
| $c_{Cov19}$ | × 2 | — | — | No |
| | ÷ 2 | -2.5 | -9.3 | Yes |
| $k_{burst}$ | × 2 | -0.0088 | +0.066 | Yes |
| | ÷ 2 | +0.0074 | +0.24 | Yes |
| $k_{Cov19}$ | × 2 | — | — | No |
| | ÷ 2 | +0.54 | +4.3 | Yes |
| $k_{AT1R}$ | +10% | — | — | No |
| | -10% | — | — | No |
| $k_{ACE2p}$ | +10% | — | — | No |
| | -10% | — | — | No |
| $k_{Ang(1-7)}$ | +10% | -0.50 | -6.7 | Yes |
| | -10% | — | — | No |
| $k_{AT2R}$ | +10% | -0.30 | -3.9 | Yes |
| | -10% | — | — | No |

## Sensitivity analysis

Many of the model parameters have not been characterized. To assess the sensitivity of model prediction on variations in those parameters, we conduct a sensitivity analysis (Table 2) in all model parameters listed in Tables 3 and 4, except for the half-life of ACE2 which has been measured experimentally [11]. For parameters whose values have not been directly measured but were inferred from experimental findings ($k_{AT1R}$, $k_{ACE2p}$, $k_{Ang(1-7)}$, and $k_{AT2R}$), we vary their baseline values by ±10%; for other parameters which are largely uncharacterized, we multiply and divide their baseline values by 2. For each of these variations, we conduct simulation of low-level SARS-CoV-2 exposure, and determine whether tissue recovery is attained, and if so, compare the peak viral load (P) and tissue damage (D) to baseline values.

Results of the sensitivity analysis are shown in Table 2. These results indicate that model results are largely insensitive to $k_{burst}$, the rate constant for release of cellular virus into plasma,

**Table 3. Predicted variables and parameters for ACE2 dynamics.** Baseline values are given for model parameters.

| Symbol | Meaning | Value |
|---|---|---|
| ACE2m | Membrane-bound ACE2 | |
| ACE2p | Plasma ACE2 | |
| ACE2i | Internalized ACE2 | |
| $P_{plasma}$ | Plasma SARS-CoV-2 | |
| $S_{ACE2m}$ | ACE2m generation rate | $1.36 \times 10^{-2}$ |
| $c_{ADAM17}$ | ACE2 shedding rate constant | $6.51 \times 10^{-5}$ |
| $c_{AT1R}$ | AT1R-mediated ACE2 internalization rate constant | $1.48 \times 10^{-6}$ |
| $c_{Cov19}$ | Rate constant for binding of SARS-CoV-2 to ACE2 | $3.00 \times 10^{-6}$ |
| $h_{ACE2}$ | ACE2 half-life | $8.5 \text{ h}^{-1}$ [11] |
| $k_{burst}$ | Rate constant for release of cellular virus into plasma | $10^{-3}$ |

**Table 4. Predicted variables and parameters for immune response.** Baseline values are given for model parameters.

| Symbol | Meaning | Value |
|---|---|---|
| P | Internalized SARS-CoV-2 | |
| N | Inflammatory agents | |
| D | Tissue damage | |
| $C_A$ | anti-inflammatory mediators | |
| $k_{Cov19}$ | SARS-CoV-2 + ACE2 entry rate scaling | $10^3$ |
| $k_{AT1R}$ | Inflammatory response to AT1R-bound Ang II | $3.50\times10^{-3}$ [28] |
| $k_{ACE2p}$ | Anti-inflammatory response to ACE2p | $2.10\times10^{-2}$ [29] |
| $k_{Ang(1-7)}$ | Anti-inflammatory response to AT2R-bound Ang (1-7) | $1.00\times10^{-2}$ [30] |
| $k_{AT2R}$ | Anti-inflammatory response to AT2R-bound Ang II | $5.10\times10^{-3}$ [31] |
| $k_{dn}$ | Maximum rate of damage produced by activated phagocytes | 0.35 |
| $\mu_d$ | Recovery rate constant for damaged tissue | 0.02 [32] |

but for all other parameters, a sufficiently large deviation from baseline values in at least one direction would render tissue recovery impossible. In particular, this analysis indicates that model results are particularly sensitive to the pro-inflammatory effect of AT1R-bound Ang II ($k_{AT1R}$) and to the anti-inflammatory effect of ACE2 ($k_{ACE2p}$). Interestingly, a sufficiently large deviation in these parameters from their baseline values in either direction would result in sustained tissue damage. This result highlights the role of inflammation as a double-edged sword. A weakened in the inflammatory response simultaneously attenuates tissue damage and impedes the system's defense against pathogens. With the invasion of a new virus, attenuation of the inflammatory defense may yield sustained tissue damage.

## Discussion

The principal goal of this study is to gain insights into the implications of hypertension and some of its medications in the pathogenesis of COVID-19. To that end, we have developed a mathematical model that represents the RAS, the binding of a key RAS component, ACE2, with SARS-CoV-2 and the subsequent cell entry, and the host's inflammatory response. The model can simulate different levels of SARS-CoV-2 exposure, and the effect of commonly prescribed anti-hypertensive medications, ACEi and ARBs, and predict tissue damage and recovery.

*What are the factors that determine the severity of COVID-19?* Our simulation results indicate that, not surprisingly, the level of exposure to SARS-CoV-2 is a key factor, as is the robustness of one's immune system (see Fig 2). Aspects of the immune system is known to be impaired in the aged population [12], which may explain, in part, the higher COVID-19 mortality in that subpopulation. Pre-existing conditions such as hypertension are also known play a role. Hypertension is a highly complex disease that is difficult to simulate. Instead, we focus on the low-grade inflammation that is frequently associated with hypertension. Not surprisingly, our simulations indicate that a pre-existing chronic inflammation may increase the difficulty of recovering from tissue damaged sustained in COVID-19 (Table 1 and Fig 3). Chronic inflammation is also found in diabetes, cardiovascular diseases, and allergies. In a recent study, elevated extracellular traps from neutrophils have been observed in COVID-19 patients and correlates with worse outcomes [13].

SARS-CoV-2 gains entry into host cells by first binding with a membrane receptor ACE2, an essential component of the RAS. This mechanism may have implications on the treatment of hypertension. Although neither ACEi nor ARBs directly affects ACE2 activities, studies in rodents have shown that these agents upregulate the expression and activity of ACE2 in cardiac

tissue [7]. This observation has prompted concerns about the potential enhanced susceptibility of patients receiving these drugs to SARS-CoV-2 infection and COVID-19 severity. *How might ACEi and ARBs impact recovery from tissue damage in COVID-19*? The answer is, it depends, as summarized in Table 1 and Fig 3:

- For individuals with only mild chronic inflammation or none at all, the administration of ACEi may indeed exacerbate tissue damage. The elevated viral entry outweighs the pro-inflammatory benefits of the drug. The caveat is that this balance depends on our baseline assumption of ACEI-mediated ACE2 upregulation: a 5-fold increase, based on a study in the rat cardiac muscle [7]. If ACEi increases ACE2 expression in human pulmonary tissue to a lesser extent or not at all, then ACEi may not impede tissue recovery.

- For the same cohort (mild or no chronic inflammation), administration of ARBs inappropriately disrupts the body's inflammatory and immune response, and is predicted to worsen tissue damage. A caveat is that how much ARBs affect the pro- and anti-inflammatory signals is not well quantified.

- For individuals with more severe inflammation, ACEi and ARBs may facilitate tissue recovery. This is attributable to the anti-inflammatory benefits of these drugs.

ACEi and ARBs are prescribed for a range of diseases including hypertension, proteinuric chronic kidney disease, and heart failure. Although other drug classes may be available for the management of hypertension, patients with proteinuric chronic kidney disease or heart failure with impaired left ventricular function have limited alternatives. Given the renal and cardiovascular protective effects of these drugs, discontinuing their use will likely have significant negative health impacts. Indeed, while some have advocated for discontinuing these medications during the pandemic, expert opinions recommended against discontinuation, given the lack of strong evidence, or evidence indicating the lack of association of ACEi or ARB treatment with increased risk for acquiring COVID-19 or for poorer outcomes [14]. Therefore, a decision regardless the prescription and usage of ACEi and ARB should not be made lightly. It is our hope that the findings of this study serve as a step in the development of appropriate and more comprehensive guidelines for the prescription of these drugs in the current and future coronavirus pandemics.

This study presents the first mechanistic model that simulates the interactions between the two of the primary players in SARS-CoV-2 infection: the RAS and the acute inflammatory response. The model allows us to weigh the anti-inflammatory benefit of ACE2 against its role in facilitating viral entry into host cells, which is an important consideration in COVID-19 pathogenesis. Despite its contributions, major limitations of the model must be acknowledged. Compared to our published RAS model [10], the present model contains a simplified representation of the RAS that assumes a constant plasma renin activity (PRA), which implies that the feedback from AT1R-bound Ang II to PRA is not represented. Thus, the model does not account for the feedback effects in which a drop in AT1R-bound Ang II, following the administration ACEi and ARBs, increases renin and Ang I, and subsequently increases Ang (1-7).

Additionally, the model only represents the acute inflammatory response; the adaptive or chronic immune responses are not included. During the incubation and non-severe stages, a specific adaptive immune response is required to eliminate the virus and to preclude disease progression to severe stages. Therefore, modeling the acute immune response in COVID-19 (e.g., Ref. [15]) may accelerate the development of strategies for boosting immune responses (anti-sera or pegylated IFNα) at this stage. Whether SARS-CoV-2 may induce a chronic immune response, as seen in HIV infection, has yet to be determined.

A major limitation of the model is the lack of available data, both for estimating model parameters and for validating model predictions. This is unfortunately often the case with diseases without an animal model. The sensitivity analysis (Table 2) suggests that a better characterization of the pro-inflammatory effect of AT1R-bound Ang II and of the anti-inflammatory effect of ACE2 would improve the model's prediction on the extent of tissue damage resulting from COVID-19. The simulations of ACEi and ARB indicate that a better understanding of the effect of these drugs on ACE2 abundance and viral entry would yield a clearer indication of these drugs on the pathogenesis of COVID-19. Carefully measured time-courses of viral titer and the concentrations of various immune agents would be useful in the development of a model with a more comprehensive immune response and in validating its predictions.

An important message of this study is that there is no one-size-fits-all approach in drug prescription. Whether ACEi and ARBs might improve or impede tissue recovery in COVID-19 depends on the conditions of the patient. It is noteworthy that significant sex differences have been reported in the RAS, and the current RAS model parameters are taken for male [10]. We conducted simulates using parameters fitted for female, and model predictions are qualitatively similar. Many other individual differences exist in hypertension besides sex [16–20], including race [21], comorbidities, as well as other medications, including metformin and SGLT2 inhibitors [22, 23] for diabetes, and these factors should be taken into account.

## Methods

### Mathematical model

The model represents the RAS and the key role of one of its components, ACE2, in the host cell entry of SARS-CoV-2, as well as the inflammatory response that it triggers. For a susceptible individual, following an initial viral exposure, the model predicts the population dynamics of SARS-CoV-2, inflammatory agents, anti-inflammatory mediators, tissue damage, ACE2, and other key RAS components. The long-term model solution predicts the severity of tissue damage.

**RAS components.** Recall that ACE2, which serves as the entry receptor for SARS-CoV and SARS-CoV-2, promotes the conversion of Ang II to Ang (1-7). The RAS model is based on our published model [10] and includes key parts of the RAS cascade; see Fig 1 (purple boxes). Ang I is converted into other forms through ACE and NEP; the respective reaction rate constants are denoted $c_{ACE}$ and $c_{NEP}$. The rate of change of [Ang I] is given by

$$\frac{d}{dt}[\text{Ang I}] = \text{PRA} - (c_{ACE} + c_{NEP})[\text{Ang I}] - \frac{\ln(2)}{h_{AngI}}[\text{Ang I}] \qquad (1)$$

where $h_{Ang\,I}$ denotes the half-life of Ang I.

Ang II is converted from Ang I through ACE and then converted into Ang (1–7) through ACE2 with the rate constant $c_{ACE2}$. Ang II also binds to the AT1R and AT2R with rate constants $c_{AT1R}$ and $c_{AT2R}$, respectively. We assume that Ang II has a half-life of $h_{Ang\,II}$. With this notation the rate of change of [Ang II] is given by

$$\frac{d}{dt}[\text{Ang II}] = c_{ACE}[\text{Ang I}] - k_{ACE2}[\text{ACE2m}][\text{Ang II}] - (c_{AT1R} + c_{AT2R})[\text{Ang II}]$$
$$- \frac{\ln(2)}{h_{AngII}}[\text{Ang II}] \qquad (2)$$

Ang II binds to AT1R with rate constant $c_{AT1R}$ to produce AT1R- bound Ang II. AT1R-bound Ang II decays with a half-life of $h_{AT1R}$, giving the rate of change of [AT1R-bound Ang

II] as

$$\frac{d}{dt}[\text{AT1R} \cdot \text{Ang II}] = c_{\text{AT1R}}[\text{Ang II}] - \frac{\ln(2)}{h_{\text{AT1R}}}[\text{AT1R} \cdot \text{Ang II}] \tag{3}$$

Ang II binds to the AT2R with rate constant $c_{\text{AT2R}}$ to produce AT2R-bound Ang II. AT2R-bound Ang II decays with a half-life of $h_{\text{AT2R}}$, giving the rate of change of [AT2R-bound Ang II] as

$$\frac{d}{dt}[\text{AT1R} \cdot \text{Ang II}] = c_{\text{AT2R}}[\text{Ang II}] - \frac{\ln(2)}{h_{\text{AT2R}}}[\text{AT2R} \cdot \text{Ang II}] \tag{4}$$

Ang (1-7) is converted by NEP from Ang I and by ACE2 from Ang II, and it decays with a half life of $h_{\text{Ang(1-7)}}$. Thus, the rate of change of [Ang (1-7)] is

$$\frac{d}{dt}[\text{Ang(1-7)}] = c_{\text{NEP}}[\text{Ang I}] + k_{\text{ACE2}}[\text{ACE2m}][\text{Ang II}] - \frac{\ln(2)}{h_{\text{Ang(1-7)}}}[\text{Ang(1-7)}] \tag{5}$$

**ACE2 dynamics.** ACE2 is a type 1 transmembrane protein (N-terminus outside, C-terminus intracellular), predominantly localized on endothelial cells where its catalytic site is exposed (the so-called "ectoenzyme") to circulating vasoactive peptides [24]. The expression level of ACE2 can be modulated through its cleavage from the cell membrane. This cleavage or shedding releases the catalytically active ectodomain and is mediated by a disintegrin and metalloprotease (ADAM 17) [25]. The ACE2 that has shedded into plasma may bind to SARS-Co-V-2 and essentially deactivate it.

Membrane-bound ACE2 (denoted ACE2m) is generated at a rate of $S_{\text{ACE2m}}$. It is cleaved by ADAM 17 and sheds with rate constant $c_{\text{ADAM17}}$ and internalized by AT1R-bound Ang II with rate constant $c_{\text{AT1R}}$. Additionally, ACE2m binds to SARS-CoV-2 with rate constant $c_{\text{Cov19}}$. Thus, the rate of change of ACE2m is given by

$$\frac{d}{dt}[\text{ACE2m}] = S_{\text{ACE2m}}$$
$$- \left( c_{\text{ADAM17}} + c_{\text{AT1R}}[\text{AT1R} \cdot \text{Ang II}] + \frac{c_{\text{Cov19}}P_{\text{plasma}}}{K_{m,\text{ACE2i}} + P_{\text{plasma}} + c_{\text{ACE2i}}[\text{ACE2p}]} \right)[\text{ACE2m}] \tag{6}$$
$$- \frac{\ln(2)}{h_{\text{ACE2}}}[\text{ACE2m}]$$

where $h_{\text{ACE2}}$ denotes the half-life of ACE2.

The shedding of ACE2m generates plasma ACE2 (denoted ACE2p).

$$\frac{d}{dt}[\text{ACE2p}] = c_{\text{ADAM17}}[\text{ACE2m}] - \frac{\ln(2)}{h_{\text{ACE2}}}[\text{ACE2p}] \tag{7}$$

The AT1R-bound Ang II-mediated internalization of ACE2m and generates ACE2i, as does the entry of the ACE2-bound SARS-CoV-2:

$$\frac{d}{dt}[\text{ACE2i}] = \left( c_{\text{AT1R}}[\text{AT1R} \cdot \text{AngII}] + \frac{c_{\text{Cov19}}P_{\text{plasma}}}{K_{m,\text{ACE2i}} + P_{\text{plasma}} + c_{\text{ACE2i}}[\text{ACE2p}]} \right)[\text{ACE2m}]$$
$$- \frac{\ln(2)}{h_{\text{ACE2}}}[\text{ACE2i}] \tag{8}$$

We also track plasma SARS-CoV-2. Its population ($P_{plasma}$) is determined by the release of internalized SARS-CoV-2 (P in Eq 10) into plasma, characterized by rate constant $k_{burst}$, and the binding of plasma SARS-CoV-2 with ACE2m and ACEp, the latter characterized by rate constant $k_{ACE2-P}$:

$$\frac{d}{dt}P_{plasma} = k_{burst}P - k_{ACE2-P}[ACE2p]P_{plasma} - \frac{c_{Cov19}P_{plasma}[ACE2m]}{K_{m,ACE2i} + P_{plasma} + c_{ACE2i}[ACE2p]} \tag{9}$$

**Inflammatory response components.** Pathogens induce an innate immune response, through which the organism aims to eliminate threats to survival and to promote tissue repair and healing. We have extended the model in [26] to represent its interactions with the RAS and the internalization of SARS-CoV-2; see Fig 1 (orange boxes). It represents the interactions among internalized (i.e., cellular) SARS-CoV-2 (denoted P), inflammatory agents ($N^*$), tissue damage (D), and anti-inflammatory mediators ($C_A$). The internalized SARS-CoV-2 population depends on its cell entry (first time in Eq 10), with rate constant $c_{Cov19}$ and scaling constant $k_{Cov19}$, a logistic term with growth rate $k_{pg}$ and carrying capacity $P_\infty$ (second term), local immune response (third term, see derivation in [26]), inhibitory action of the activated phagocytes (fourth term), and the release of internalized viral content into plasma following cell eruption (five term):

$$\frac{d}{dt}P = k_{Cov19}\left(\frac{c_{Cov19}P_{plasma}}{K_{m,ACE2i} + P_{plasma} + c_{ACE2i}[ACE2p]}\right)[ACE2m] + k_{pg}P\left(1 - \frac{P}{P_\infty}\right) - \frac{k_{pm}s_m P}{\mu_m + k_{mp}P}$$
$$- k_{pn}f(N^*) - k_{burst}P \tag{10}$$

where

$$f(V) = \frac{V}{1 + (C_A/c_\infty)^2}$$

describes the inhibitory action of the anti-inflammatory agents. To simulate the delay of immune response, $k_{pm}$ and $k_{pn}$ are initialized to 10% of their baseline value while $P < P_c$, where the critical $P_c$ is taken to be 2 pathogen units.

Inflammatory agents such as phagocytes are activated in response to damaged tissue and pathogens, as well as its own positive feedback loop (first term in Eq 11). Additionally, AT1R-bound Ang II promotes inflammation (second term). $N^*$ decays with a rate constant $\mu_n$.

$$\frac{d}{dt}N^* = \frac{s_{nr}R}{\mu_{nr} + R} + k_{AT1R}[AT1R \cdot Ang\ II] - \mu_n N^* \tag{11}$$

where

$$R = f(k_{nn}N^* + k_{np}P + k_{nd}D)$$

Tissue damage is enhanced by the activated phagocytes but attenuated by the anti-inflammatory agents; it also decays with a rate constant $\mu_d$.

$$\frac{d}{dt}D = k_{dn}f_s(f(N^*)) - \mu_d D \tag{12}$$

where

$$f_s(V) = \frac{V^6}{x_{dn}^6 + V^6}$$

The $C_A$ equation contains a source of $C_A$, denoted $s_c$, and a term modeling the production of anti-inflammatory mediator from damage and activated phagocytes, which takes the form $\frac{k_{cn}Q}{1+Q}$ and is inhibited by $C_A$. This expression is a Michaelis-Menten-type term, in which $k_{cnd}$ controls the effectiveness of damage, relative to activated phagocytes, in producing $C_A$. The model also represents the effect of anti-inflammatory agents: plasma ACE2, Ang (1-7) and AT2R-bound Ang II, as follows

$$\frac{d}{dt}C_A = s_c + \frac{k_{cn}Q}{1+Q} + k_{\text{ACE2p}}[\text{ACE2p}] + k_{\text{Ang}(1-7)}[\text{Ang}(1-7)] + k_{\text{AT2R}}[\text{AT2R} \cdot \text{Ang II}]$$
$$- \mu_c C_A \tag{13}$$

where $\mu_c$ denotes the decay rate constant and

$$Q = f(N^* + k_{cnd}D)$$

## Model parameters

Parameters for the RAS are taken from the male model in [10]. Parameters describing ACE2 dynamics are given in Table 3. $S_{\text{ACE2m}}$, $c_{\text{ADAM17}}$, and $k_{\text{AT1R}}$ are estimated by setting Eqs 6–9 to steady state, assuming a steady-state distribution of ACE2m : ACE2p : ACE2i of 0.94:0.045:0.015. $c_{\text{Cov19}}$ is chosen to fit the viral growth rate reported in [27]. Parameters for the innate immune response are taken from Ref. [26]. Additional parameters are shown in Table 4.

**Initial conditions.** RAS variables are initialized to their steady-state values in the corresponding standalone RAS model, i.e., with fixed ACE2 expression. Similarly, the immune response variables are initialized to their steady-state values in a standalone immune response model with no pathogen (i.e., P = 0).

## Author Contributions

**Conceptualization:** Mehrshad Sadria, Anita T. Layton.

**Data curation:** Mehrshad Sadria, Anita T. Layton.

**Formal analysis:** Mehrshad Sadria, Anita T. Layton.

**Funding acquisition:** Anita T. Layton.

**Investigation:** Mehrshad Sadria, Anita T. Layton.

**Methodology:** Mehrshad Sadria, Mehrshad Sadria, Anita T. Layton, Anita T. Layton.

**Resources:** Anita T. Layton.

**Software:** Mehrshad Sadria, Anita T. Layton.

**Validation:** Mehrshad Sadria, Anita T. Layton.

**Visualization:** Mehrshad Sadria, Anita T. Layton.

**Writing – original draft:** Mehrshad Sadria, Anita T. Layton.

**Writing – review & editing:** Mehrshad Sadria, Anita T. Layton.

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
