## [Decision Letter · Decision Letter 0]

2 May 2020

Dear Dr. Layton,

Thank you very much for submitting your manuscript "Use of Angiotensin-Converting Enzyme Inhibitors and Angiotensin Receptor Blockers During the COVID-19 Pandemic: A Modeling Analysis" for consideration at PLOS Computational Biology.

As with all papers reviewed by the journal, your manuscript was reviewed by members of the editorial board and by several independent reviewers. In light of the reviews (below this email), we would like to invite the resubmission of a significantly-revised version that takes into account the reviewers' comments.

The reviewers, in particular reviewer 2, offer a number of major criticisms for improving the validity and potential impact of this work. Aiming to balance expediting research relevant to covid-19 with the need for judicious validation, I hope that it will be possible to satisfy the major concerns raised by these reviewers. Furthermore, I want to point out Reviewer 2 makes the important point that with adherence to antihypertensive medications already not great among some patients, there is potential danger in advocating for not taking one's medications. Although the intent of this paper is not ]to directly inform clinical decision making, I concur with the reviewer that discussion on this point is called for.

We cannot make any decision about publication until we have seen the revised manuscript and your response to the reviewers' comments. Your revised manuscript is also likely to be sent to reviewers for further evaluation.

Sincerely,

Daniel A Beard

Deputy Editor

PLOS Computational Biology

Reviewer's Responses to Questions

**Comments to the Authors:**

Reviewer #1: Reviewer critiques and comments are found within the uploaded word doc

Reviewer #2: attached

**Have all data underlying the figures and results presented in the manuscript been provided?**

Reviewer #1: Yes

Reviewer #2: Yes

PLOS authors have the option to publish the peer review history of their article (what does this mean?). If published, this will include your full peer review and any attached files.

Reviewer #1: Yes: Andrew D. Marquis

Reviewer #2: No
---

## [Decision Letter · Decision Letter 1]

6 Aug 2020

Dear Dr. Layton,

Thank you very much for submitting your manuscript "Use of Angiotensin-Converting Enzyme Inhibitors and Angiotensin II Receptor Blockers During the COVID-19 Pandemic: A Modeling Analysis" for consideration at PLOS Computational Biology. As with all papers reviewed by the journal, your manuscript was reviewed by members of the editorial board and by several independent reviewers. The reviewers appreciated the attention to an important topic. Based on the reviews, we are likely to accept this manuscript for publication, providing that you modify the manuscript according to the review recommendations.

Sincerely,

Daniel A Beard

Deputy Editor

PLOS Computational Biology

Daniel Beard

Deputy Editor

PLOS Computational Biology

[LINK]

Reviewer's Responses to Questions

**Comments to the Authors:**

Reviewer #1: Reviewer comments are uploaded as a word doc.

Reviewer #2: The manuscript is much improved. The reviewer appreciates the addition of the sensitivity analysis in Table 2. However, I have a few requests, mostly minor except one, for improving this table and its explanation in the text.

1. There are a lot of symbols and abbreviations in this manuscript. A list of all symbols and their definitions would really help the reader.

2. Even if you make a list, it would be helpful to state in the Table caption (or header row) of Table 2 that P and D refer to peak viral load and tissue damage.

3. What do the dashed lines mean? No change? Why not just put 0%?

4. I don’t fully understand the table. For instance, for S_ACE2m, dividing it by 2 increases tissue damage 133%, but also leads to recovery? This needs to be explained a little more.

5. You say that “in particular, this analysis indicates that model results are particularly sensitive to the pro-inflammatory effect of AT1R-bound Ang II (AT1R) and to the anti-inflammatory effect of ACE2 (ACE2p).”

I can’t tell this from the table. Their values for delta P and delta D have dashes, so it is not clear what the sensitivity is.

6. You continue that “Interestingly, a sufficiently large deviation in these parameters from their baseline values in either direction would result in sustained tissue damage.” This seems really unintuitive and difficult to understand. Why would increasing the anti-inflammaotry effect of ACE2 (+10%) make recovery LESS likely? Why would decreasing the inflammatory effect of KAT1R make recovery less likely? These findings need to be explained further.

7. Some key parameters are missing from this table – Especilaly Kdn and ud – the rate constants in equation 12 for tissue damage. These relative rates are likely to be hugely important in determining how D changes over time, and thus whether recovery occurs. They should be added.

**Have all data underlying the figures and results presented in the manuscript been provided?**

Reviewer #1: Yes

Reviewer #2: Yes

PLOS authors have the option to publish the peer review history of their article (what does this mean?). If published, this will include your full peer review and any attached files.

Reviewer #1: **Yes: **Andrew D. Marquis

Reviewer #2: No
---

## [Editor Report · Decision Letter 2]

10 Aug 2020

Dear Dr. Layton,

We are pleased to inform you that your manuscript 'Use of Angiotensin-Converting Enzyme Inhibitors and Angiotensin II Receptor Blockers During the COVID-19 Pandemic: A Modeling Analysis' has been provisionally accepted for publication in PLOS Computational Biology.

Best regards,

Daniel A Beard

Deputy Editor

PLOS Computational Biology

Daniel Beard

Deputy Editor

PLOS Computational Biology

---

## [Editor Report · Acceptance letter]

23 Sep 2020

PCOMPBIOL-D-20-00606R2 

Use of Angiotensin-Converting Enzyme Inhibitors and Angiotensin II Receptor Blockers During the COVID-19 Pandemic: A Modeling Analysis

Dear Dr Layton,

I am pleased to inform you that your manuscript has been formally accepted for publication in PLOS Computational Biology. Your manuscript is now with our production department and you will be notified of the publication date in due course.

With kind regards,

Matt Lyles
